# communications
## engineering

# Deep learning-based approach for high spatial resolution fibre shape sensing

Samaneh Manavi Roodsari [1] [✉], Sara Freund[1], Martin Angelmahr[2], Carlo Seppi[1], Georg Rauter[1], Wolfgang Schade[2] & Philippe C. Cattin [1]

Fiber optic shape sensing is an innovative technology that has enabled remarkable advances in various navigation and tracking applications. Although the state-of-the-art fiber optic shape sensing mechanisms can provide sub-millimeter spatial resolution for off-axis strain measurement and reconstruct the sensor's shape with high tip accuracy, their overall cost is very high. The major challenge in more cost-effective fiber sensor alternatives for providing accurate shape measurement is the limited sensing resolution in detecting shape deformations. Here, we present a data-driven technique to overcome this limitation by removing strain measurement, curvature estimation, and shape reconstruction steps. We designed an end-to-end convolutional neural network that is trained to directly predict the sensor's shape based on its spectrum. Our fiber sensor is based on easy-to-fabricate eccentric fiber Bragg gratings and can be interrogated with a simple and cost-effective readout unit in the spectral domain. We demonstrate that our deep-learning model benefits from undesired bending-induced effects (e.g., cladding mode coupling and polarization), which contain high-resolution shape deformation information. These findings are the preliminary steps toward a low-cost yet accurate fiber shape sensing solution for detecting complex multi-bend deformations.

[1] Department of Biomedical Engineering, University of Basel, Hegenheimermattweg 167C, Allschwil 4123, Switzerland. [2] Department of Fiber Optical Sensor Systems, Fraunhofer Institute for Telecommunications, Heinrich Hertz Institute, HHI, Am Stollen 19H, Goslar 38640, Germany. [✉]email: samaneh.manavi@unibas.ch

Fiber optic shape sensing has proven to have great potential, especially in medical applications such as catheter navigation, surgical needle tracking, and flexible endoscope navigation. Compared to other common navigation technologies (e.g., optical trackers, electromagnetic sensors, or medical imaging), fiber shape sensing has many advantages, such as immunity to electromagnetic fields, bio-compatibility, and high flexibility. Fiber shape sensors are small in diameter, easily integrable into flexible instruments, and require no line-of-sight. Distributed sensors based on multicore fibers can also provide high-resolution shape measurements[1,2].

Fiber shape sensors measure off-axis strain, which is then used to compute the directional curvature and reconstruct the sensor's shape[3]. Various fiber sensor configurations have been investigated for off-axis strain measurement, including multicore fibers with[4–6] or without[7–9] fiber Bragg gratings (FBG) in their cores, fibers with cladding waveguide FBGs[10], and fiber bundles made from multiple single-mode fibers that contain FBG arrays[11–15]. Accurate shape reconstruction necessitates high spatial resolution in off-axis strain measurement. With a distributed fiber shape sensor, sub-millimeter spatial resolution can be achieved[1]. However, these sensors require the use of specialized and costly optical reflectometers to analyze the back-scattered light and retrieve strain variations[16–19]. Moreover, the signal-to-noise ratio of the back-scattering trace in such sensors depends on the spatial resolution and the level of applied strain. Quasi-distributed sensors, on the other hand, have more cost-effective readout unit systems (e.g., FBG interrogators). However, their spatial resolutions are limited by the low sensing plane density[4,20], making them inapplicable for tracking complex shape deformations. Therefore, there is a need for a cost-effective, high-resolution, and accurate fiber shape sensing technique.

Among cost-effective fiber shape sensors interrogated in the spectral domain, eccentric FBG (eFBG) sensors show great capacity for tracking applications, thanks to their unique sensing mechanism[21–23]. Each sensing plane in eFBG shape sensors consists of three highly localized FBGs, written off-axis in the fiber's core (also known as edge-FBG triplet), as shown in Fig. 1a[21]. Shape deformations are commonly computed from the displacement of the fundamental mode-field inside the optical fiber, estimated through spectral intensity modifications (see Fig. 1b, c)[21,22]. This approach is known as the mode-field displacement method (MFD). However, several other effects, including bending-sensitive mode coupling[24–27], polarization-dependent losses[28–32], and wavelength-dependent bending losses[33–39], also modify the spectral profile of eFBGs. These effects cannot be accurately modeled, and their impact on the sensor's spectra is indistinguishable from the mode-field displacements. Further details on the eFBG configuration, sensing mechanism, and bending-induced effects are provided in "Methods".

In this paper, we introduce an end-to-end data-driven modeling technique based on deep learning (DL) that effectively identifies meaningful patterns in the eFBG signal, even in the presence of uncontrolled bending-induced effects. By incorporating these additional sources of information, our technique considerably improves the accuracy of shape prediction. Moreover, our approach enables high spatial resolution shape estimation directly from the eFBG sensor's signal, eliminating the need for strain measurement, curvature computation, and shape reconstruction steps.

## Results and discussion

**Training and testing datasets**. The eFBG fiber sensor used in this work is 30 cm long and consists of five sensing planes separated by 5 cm from each other. At each sensing plane, three off-axis FBGs are inscribed at a radial distance of approximately 2 μm from the top, left, and right sides of the fiber's core. The dataset used for developing the DL-based model is collected using a similar setup reported in our previous work[40] (see "Methods" for more detail). We used three normalized spectral scans that were consecutively measured as input data to the proposed DL model. Each scan was recorded from 800 to 890 nm, comprising 190 wavelength components. The target data are the relative coordinates of 20 discrete points (reflective markers of the tracking system) measured over the length of the shape sensor (more detail on data preprocessing is available in[41]). This dataset consists of approximately 58,000 samples collected during 30 min of random movement of the fiber sensor. To evaluate the predictive performance of the trained model in an unbiased way, the samples were first shuffled and then split into Train-Validation-Test subsets, with 80% used for training, 10% for validating, and 10% for testing. In the remainder of this paper, we refer to this testing dataset as $Test_1$. A separate set of data, denoted as $Test_2$, consisting of approximately 5800 samples, was recorded to evaluate the performance of the trained model for unseen shapes resulting from continuous movement. Additionally, we collected 320 samples, referred to as $Test_3$, in which specific sensor regions were bent. Further details are provided in the Methods section.

**Neural network design**. The DL model needs a specially designed network architecture to extract essential features from the sensor's spectra and to accurately predict its corresponding shape. In this study, we employed an optimization algorithm inspired by the Hyperband optimizer[42] to fine-tune the network's hyperparameters. These hyperparameters, which cannot be directly determined from the training data, play a crucial role in model performance. Figure 2 illustrates the architecture of the best-performing configuration achieved after hyperparameter tuning (see "Methods" for further details).

**Shape prediction evaluation**. We evaluated the performance of the DL approach using the three testing datasets and compared it with the MFD method. It should be noted that the density of sensing planes in our eFBG shape sensor is insufficient for the MFD method to accurately estimate complex deformations. Nevertheless, we conducted this test to highlight the superiority of the proposed data-driven technique (the DL method).

Table 1 presents the shape error metrics, including the tip error, that is, the Euclidean distance between the true and the predicted coordinate of the sensor's tip and the root-mean-square of the Euclidean distance (RMSE) between the true and the predicted coordinates of the discrete points along the sensor's length. When using the $Test_1$ dataset, the MFD approach yielded a median tip error of 111.3 mm with an interquartile range (IQR) of 121.5 mm. These error values were reduced to 98.5 and 46 mm when using the $Test_2$ dataset. The performance difference can be attributed to the fact that the $Test_1$ dataset contains more diverse shapes as the samples are randomly selected from a larger dataset, whereas $Test_2$ represents continuous sensor movement over a shorter period. As expected, the error values are considerably high across all testing datasets since there is too little information available for the MFD approach to estimate complex shape deformations accurately.

The DL method, on the other hand, considerably improved the accuracy of shape prediction for $Test_1$ samples, resulting in a median tip error of 2.1 mm with an IQR of 2.6 mm. These values increased to 17.1 mm and 12.6 mm on the less diverse $Test_2$ samples. This is because the DL model can only learn to extract the most general and relevant features from the input signal when

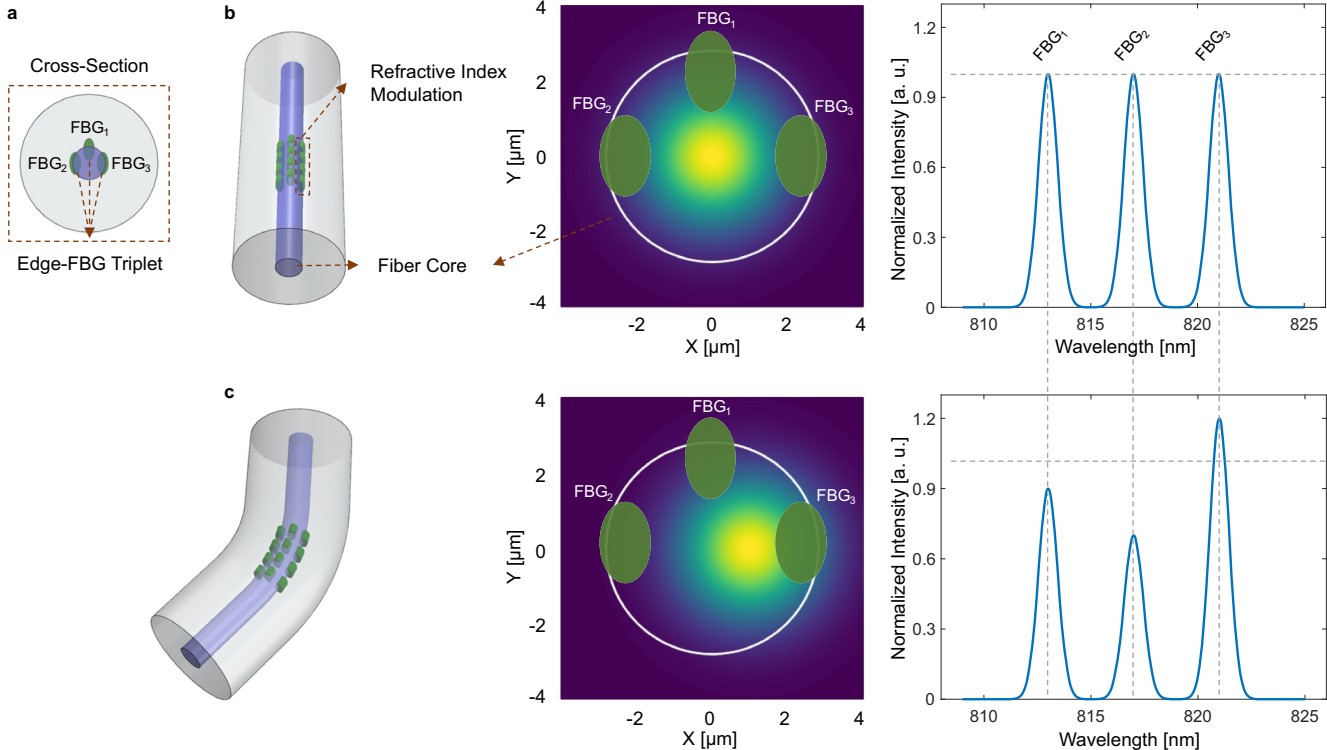

**Fig. 1 Fiber Bragg grating (FBG) configuration and working principle of the eccentric FBG (eFBG) sensor. a** Sketch of the cross-section view of the eFBG sensor. Each sensing plane of the eFBG sensor consists of three FBGs inscribed off-axis with ~90° angular separation (also known as edge-FBG triplet). **b** Mode-field distribution of a straight single-mode fiber and the expected signal from eFBGs within the same sensing plane. **c** When the fiber is curved, mode-field distribution moves in the opposite direction of bending, which affects the relative intensity between the eFBGs.

the training dataset adequately represents the expected sensor signals. However, in the case of the $Test_2$ dataset, less than 2% of the samples have at least 100 similar examples in the training data. To measure similarity, we employed a maximum RMSE threshold of 5 mm after evaluating various thresholds. This indicates that the 30 min of manual shape manipulation is insufficient to cover the full working space of the sensor and to create a representative training dataset for the model to generalize effectively. On the other hand, in the $Test_1$ dataset, almost 20% of the samples have at least 100 similar examples in the training dataset. This means that the DL method is being tested on samples that the model has already learned to handle, simulating a situation where the training dataset represents the expected sensor shapes.

The shape evaluation results of the $Test_1$ dataset define the lower performance limit for our model. Such performance difference also suggests that the DL model is better trained as application-specific, since it can focus more effectively on relevant features when learned from the expected shape distribution of the sensor. On the other hand, when training data covers a wide range of expected behaviors from the sensor, the DL model may simply "memorize" the corresponding shape for each signal without searching for relevant features in the sensor's spectrum. To investigate this further, we compared the performance of our DL method with a dictionary-based algorithm. In this approach, a pre-defined dictionary was created using all training and validation samples. The shape prediction was then made by finding the closest spectrum to the test sample and presenting its corresponding shape. This technique is equivalent to the k-nearest neighbors (kNN) algorithm with a $k$ value of 1. The median tip errors for the $Test_1$ and $Test_2$ datasets using this dictionary-based algorithm are 5.9 and 50.0 mm, with IQR values

of 3.9 and 43.3 mm, respectively. We also evaluated the kNN algorithm with $k$ values of 3, 5, 7, and 9, which resulted in median tip errors of 6.4, 7.8, 9.1, and 10.1 mm for the $Test_1$ dataset and 47.3, 46.1, 45.4, and 44.8 mm for the $Test_2$ dataset, respectively. All error values are higher compared to the errors obtained using our DL technique. This shows that our DL model generalizes well and provides more accurate shape predictions.

Two essential factors have to be considered when working with dictionaries: the size of the dictionary and the execution time required to find the best matching example. To obtain an accurate shape estimation for a given sample, the dictionary should contain a sufficiently large number of stored samples to cover all possible examples, which leads to a long execution time. Thus, there is a trade-off between accuracy and execution time when using this approach. However, extensive training data do not negatively affect the inference time in the DL method, as the resulting model size is independent of the training data size. This makes the DL method advantageous in terms of both accuracy and efficiency.

Our observations showed that the designed DL model can accurately recognize deformations even when they occur between the sensing planes. To further investigate this intriguing finding, we evaluated the shape predictions using the $Test_3$ dataset, in which the deformations are exclusively applied between the sensing planes. The $Test_3$ dataset comprises four deformation examples, each repeated twice and measured 40 times. As anticipated, the classical MFD method was not able to accurately predict the sensor's shape for such deformations, as the deformed area was not at any of the sensing planes. In contrast, when using the DL method, we achieved a median tip error of 6 mm, which is approximately six times smaller than the median tip error obtained using MFD on this dataset. The precision of the predicted tip position in the $Test_3$ dataset averages at 1.9 mm.

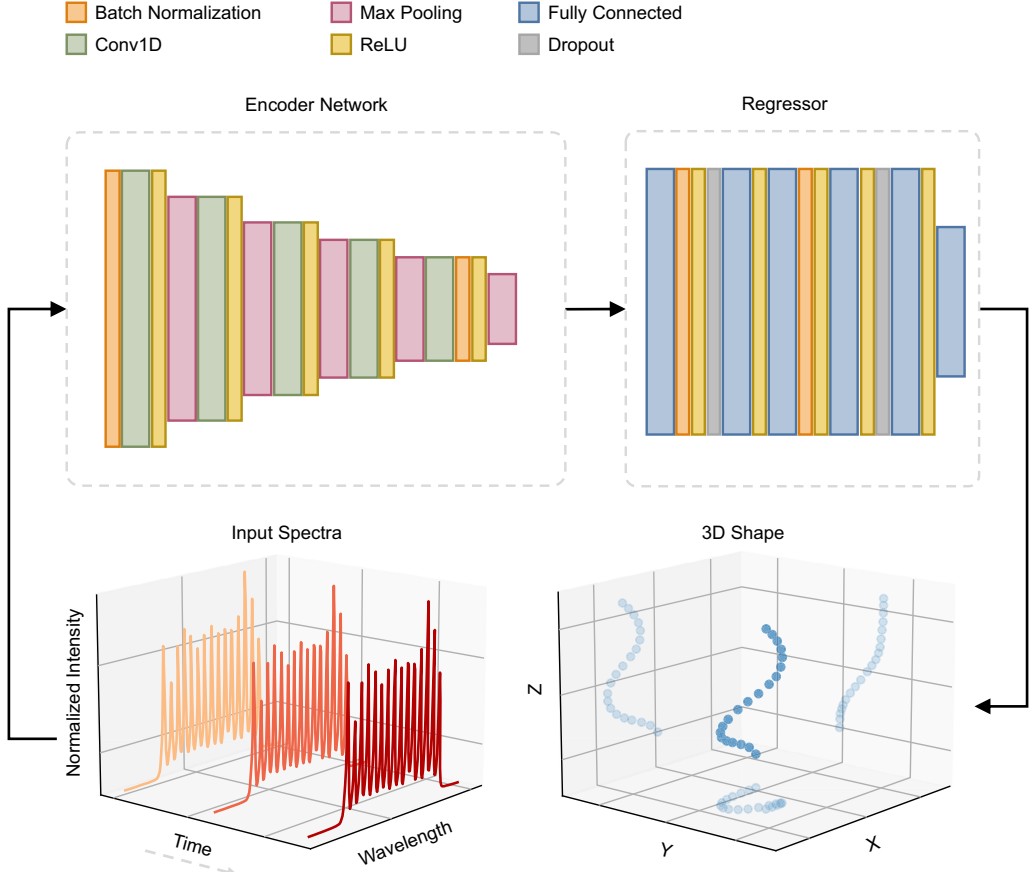

**Fig. 2 Architecture of the best-performing configuration after hyperparameter tuning.** The architecture includes five 1D convolutional layers (Conv1D), six fully connected layers, five max pooling layers, four batch normalization steps, and two dropout steps. The designed network receives three consecutive spectral scans as the input and predicts the relative coordinates of 20 discrete points over the sensor's curve. More details on the channel, kernel, and pooling sizes are available under "Methods". ReLU rectified linear unit.

**Table 1 Shape evaluation errors in mode-field displacement (MFD) and deep-learning (DL) methods using test sets $Test_1$, $Test_2$, and $Test_3$.**

| Dataset | Method | Tip error [mm] | | RMSE [mm] | |
|---|---|---|---|---|---|
| | | Median | IQR | Median | IQR |
| $Test_1$ | MFD | 111.3 | 121.5 | 59.4 | 71.7 |
| | DL | 2.1 | 2.6 | 1.5 | 1.6 |
| $Test_2$ | MFD | 98.5 | 46.0 | 53.8 | 29.1 |
| | DL | 17.1 | 12.6 | 9.8 | 7.0 |
| $Test_3$ | MFD | 39.5 | 34.7 | 17.1 | 18.3 |
| | DL | 6.0 | 9.0 | 5.1 | 6.6 |

RMSE root-mean-square error, IQR interquartile range.
footnote It's important to clarify that the MFD method is not intended to represent the current state-of-the-art technology. Instead, it serves as a baseline to illustrate what can be achieved through analytical methods when utilizing a low sensing density in eFBG fiber sensors.

An example from the $Test_3$ samples, where the sensor experienced bending between the sensing planes 3 and 4, is depicted in Fig. 3a. It is important to note that the intensity ratio of the eFBG Bragg peaks in each sensing plane can also be influenced by various factors, apart from fundamental mode-field displacements, as previously mentioned. The MFD approach, however, does not consider such effects and is thus incapable of correctly interpreting the resulting signal variations. In contrast, the DL model managed to accurately predict the sensor's shape by considering the full spectral profile, including the minute changes occurring at wavelengths outside the Bragg

resonances. Figure 3b illustrates the finite difference analysis of the loss value with respect to the 190 wavelength components of the input spectra. A higher difference indicates the greater importance of the corresponding wavelength component for shape prediction in this example. This difference provides an influence evaluation for each wavelength component of the input spectra to decode the model's predictions (see "Methods" for detailed information). Figure 3c provides a deeper insight into this analysis. For all 190 wavelength components, the Euclidean distance between the predicted relative coordinates of each marker before and after the spectral modification is depicted through a color map. The contribution of each wavelength component to the relative coordinate prediction of all 20 markers can be discerned from the presented color map in Fig. 3c.

Another important finding of our study is the DL model's ability to detect deformations occurring after the last sensing plane. Figure 4a illustrates an example in which a 3 cm long segment, 1 cm after the last sensing plane, was deformed. Similar to the example depicted in Fig. 3, the MFD method was not able to predict the sensor's shape in such deformations. In contrast, the DL model employed relevant features in the side slopes of the eFBG spectra to predict the correct shape (see Fig. 4b, c). This intriguing performance can be attributed to the wavelength-dependent interference between the back-reflected light from the air-glass interface at the fiber's end tip (Fresnel reflection) and the incident downstream light occurring in the region after the last sensing plane. Deformations in this region impact interferences in two ways: first, the bending induces changes in the spectral profile of

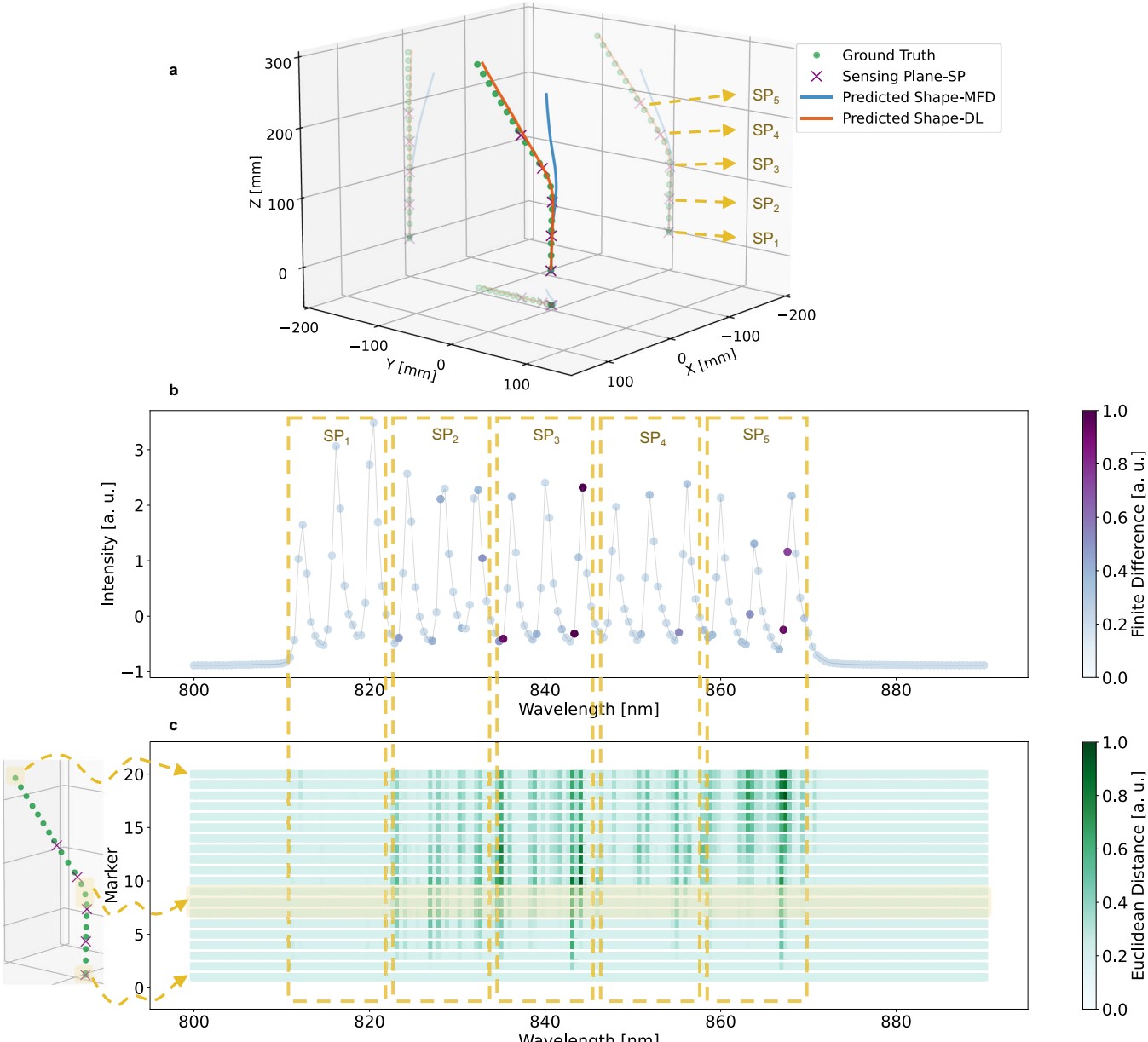

**Fig. 3 Decoding the deep-learning (DL) model decision for deformations between sensing planes. a** Example from the *Test*₃ dataset demonstrating the bending of the sensor between the sensing planes 3 and 4. The true shape (ground truth) is indicated by green circles. The five sensing planes of the sensor are shown with × signs. The predicted shapes using the mode-field displacement method (MFD) and the DL method are shown with blue and orange solid lines, respectively. **b** Visualization of the finite difference of the loss value with respect to the input spectral elements. Wavelength components shown with colors closer to dark purple contribute more to the model's decision in this particular example. **c** Highlighting the importance of input spectral elements in the relative coordinate prediction of all 20 markers based on the magnitude of the Euclidean distance between the predicted relative coordinates of each marker before and after spectral modification. Each row corresponds to one marker, and the color map represents the importance of the wavelength component. Wavelength components shown with colors closer to dark green have a greater impact on the model's decision. The markers at the bent area are highlighted in the presented color map. SP$_i$ $i_{th}$ Sensing Plane.

the downstream light, and second, it alters the coupling conditions between the back-reflected and the downstream lights. As a result, the measured spectra from the fiber sensor exhibit small variations, reflecting the influence of deformations on the interference pattern. More examples of the sensor's predicted shapes using the DL and the MFD methods on datasets *Test*₁, *Test*₂, and *Test*₃ are provided in Supplementary Movies 1–3, respectively.

**Optimum number of sensing planes**. A key factor in eFBG sensors when employing the MFD method is the number of

sensing planes for detecting shape deformations. As with any other quasi-distributed shape sensor, the spacing between the sensing planes determines the sensor's spatial resolution in shape measurements. When dealing with complex shape deformations, a limited number of sensing planes (resulting in low spatial resolution) can lead to large tip errors in methods that involve shape reconstruction (e.g., the MFD method). In this section, we present a theoretical analysis to determine the minimum number of sensing planes required in eFBG sensors when employing the MFD method to achieve the same level of shape prediction accuracy as attained by our DL method using five sensing planes.

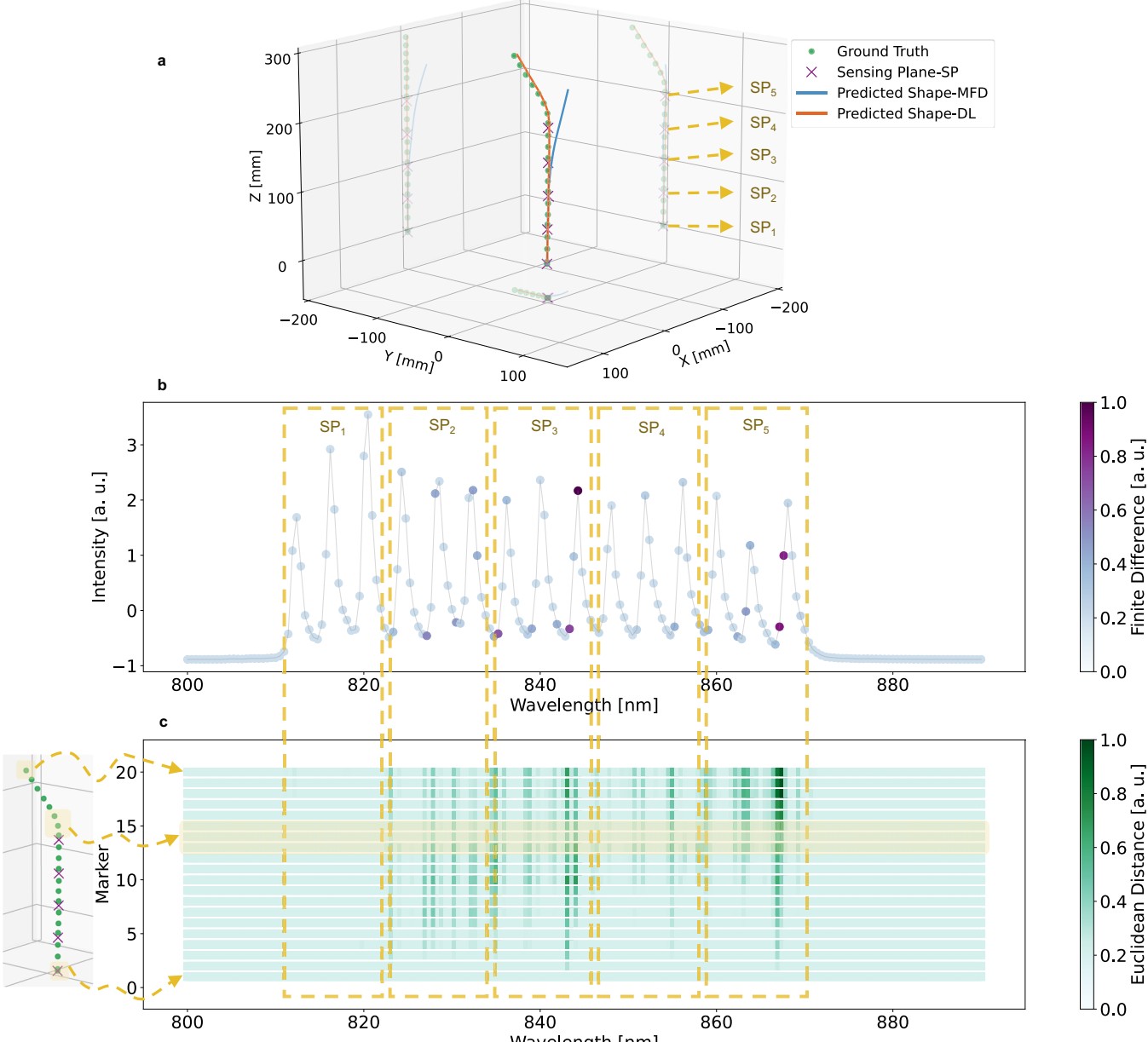

**Fig. 4 Decoding the deep-learning (DL) model decision for deformations after the last sensing plane. a** Example from the $Test_3$ dataset in which a 3 cm long segment, 1 cm after the last sensing plane, is deformed. **b** Visualization of the finite difference of the loss value with respect to the input spectral elements. Wavelength components shown with colors closer to dark purple contribute more to the model's decision in this particular example.
**c** Highlighting the importance of input spectral elements in the relative coordinate prediction of all 20 markers based on the magnitude of the Euclidean distance between the predicted relative coordinates of each marker before and after spectral modification. Each row corresponds to one marker, and the color map represents the importance of the wavelength component. Wavelength components shown with colors closer to dark green have a greater impact on the model's decision. The markers at the bent area are highlighted in the presented color map. MFD mode-field displacement. SP$_i$ $i_{th}$ Sensing Plane.

In this theoretical analysis, we simulated the shape reconstruction error for different spatial resolutions. Our methodology involves interpolating the discrete curve points along the sensor's true shape, measured by the motion capture system, using a Spline with a resolution of 0.1 mm (this value was chosen empirically). Subsequently, we compute the curvature and torsion —representing the curve's deviation from the osculating plane— at the query points. By utilizing the computed curvatures and bending directions at the sensing planes, we reconstructed the spatial curve and compared it with the true shape.

For a 25 cm long sensor with 50 mm spatial resolution (equivalent to five sensing planes), the median tip error of the

reconstructed shapes, evaluated using the $Test_1$ and $Test_2$ datasets, is approximately 50 mm. This error is nearly 16 times higher compared to the performance achieved by the DL approach (see Table 1). In order to achieve a median tip error of 3 mm, a similar spatial resolution is necessary, implying that the MFD method would require approximately 84 sensing planes consisting of 252 eFBGs.

## Conclusion

In this paper, we developed a fiber shape sensing mechanism with a data-driven technique, eliminating the need for off-axis strain

measurement and curvature computation at discrete points along the fiber sensor to estimate its 3D shape. Our approach utilizes an easy-to-fabricate eFBG sensor combined with a simple and cost-effective readout unit. We designed an end-to-end DL algorithm that can learn directly from the sensor's signal to predict its corresponding shape. We extensively evaluated the shape prediction accuracy of our designed model (the DL method) in various testing conditions and compared it with an exemplary experiment, the MFD method. Our findings highlight that the spatial resolution of off-axis strain measurement in FBG-based (quasi-distributed) shape sensors is the main limitation, as the deformations between the sensing planes are not detected in complex shapes. However, our DL method compensates for this limitation by utilizing the full spectrum of our eFBG sensor, including the Bragg resonance's side slopes, to predict complex shape deformations.

We believe that the DL model exploits the impact of bending-induced phenomena, including cladding mode coupling, bending-loss oscillations, and polarization-dependent losses, as additional sources of information to overcome the spatial resolution limitation for detecting complex deformations. As a result, there is no need to modify the fiber sensor design or its interrogation system to mitigate the impact of these bending-induced phenomena. Our developed DL method considerably reduces the shape prediction error for 3D curves within a curvature range of 0.58–33.5 m$^{-1}$, achieving a reduction factor of approximately 50 compared to the MFD method. Moreover, we demonstrated that the designed DL model generalizes nicely, as its performance surpasses that of a dictionary-based algorithm by a factor of two. Importantly, our proposed shape sensing solution offers a cost-effective alternative, being 30 times less expensive than commercially available distributed fiber shape sensors while maintaining a similar level of accuracy.

In summary, our research presents a promising approach to fiber shape sensing by combining an easy-to-fabricate eFBG sensor, a data-driven DL model, and the exploitation of bending-induced phenomena. We believe that this work has the potential to drive advancements in efficient and cost-effective shape sensing across various applications.

## Methods

**Working principle of eFBG sensor**. When the eFBG sensor undergoes bending, the field distribution of the fundamental mode shifts away from the center of the fiber core[21–23] (see Fig. 1b). Displacements of the mode-field's centroid lead to intensity changes in the reflected signal from the eFBGs[21]. From the intensity ratio between the eFBGs at each sensing plane, the directional curvature is computed and interpolated at small arc elements to reconstruct the 3D shape of the sensor[21]. For the sake of simplicity, this approach assumes that no other physical phenomena occurring inside a bent optical fiber influence the intensity ratio between the eFBGs within the same sensing plane.

However, positioning FBGs away from the core axis breaks the cylindrical symmetry of the fiber, which increases coupling from the core mode to the cladding modes[24,25]. The strength of this mode coupling varies when the fiber is bent, as it affects the overlap integral between the interacting modes[24,26]. Bending an optical fiber causes strain-induced refractive index changes and displaces the intensity distribution of the propagating light[22,43], which directly influences the coupling efficiency. Therefore, the intensity of the cladding modes changes when the fiber is bent. In eFBGs, the formation of cladding-mode resonances in fiber gratings enables highly sensitive full-directional bending response through simple light intensity measurements[27]. Although cladding modes are typically stronger in stripped fibers or fibers with lower refractive index coatings than the cladding layer[24,25], they have also been observed in standard fibers coated with higher refractive index materials[44]. Any recoupling between the excited cladding resonances and the fundamental mode affects the relative intensity values between the eFBGs.

FBG interrogators used for quasi-distributed sensors typically consist of a broadband light source (e.g., super luminescent diode (SLED)) and a grating-based spectrometer. The emitted light from SLEDs is partially polarized, meaning that it undergoes wavelength-dependent polarization changes[28] when propagating through a birefringence medium, such as a bent fiber[29–32]. Additionally, the efficiency of the spectrometer grating is sensitive to polarization, leading to polarization-dependent losses that affect the spectral profile. Consequently, the measured intensity ratio between the Bragg peaks is modified. The impact of polarization in intensity-based fiber sensors is often mitigated by using a polarization scrambler to randomize the polarization state or by employing polarization-insensitive spectroscopy instruments.

It is well known that light power loss increases when optical fibers bend[33,45]. This bending loss is typically observed as spectral modulations caused by coherent coupling between the core mode and the radiated field reflected by the cladding-coating and the coating-air interfaces (commonly referred to as whispering gallery modes)[34,46]. The reflected field at the coating-air boundary causes short-period modulations due to the longer re-injection path[34,46], while reflections at the closer cladding-coating interface cause long-period resonances[35–37,46]. It is important to note that these bending-induced attenuation losses are also influenced by temperature variations. Temperature changes affect the refractive index of the coating layer, thereby influencing the coupling between the core and the cladding whispering gallery modes[38]. Several models have been proposed to evaluate the peak positions and shapes of bending losses[35,36,39]. The strong wavelength dependence of bending losses further complicates the design of intensity-based sensors[46] as it modulates the spectral profile and affects the intensity ratio at the Bragg peaks of the eFBGs within the same sensing plane.

**Setup**. The data acquisition setup used for developing the DL-based model is depicted in Fig. 5. We used a cost-effective FBG interrogator (MIOPAS GmbH, Goslar, Germany) consisting of an uncooled transmit optical sub-assembly (TOSA) SLED module and a near-infrared (NIR) micro-spectrometer with a resolution of 0.5 nm. This setup allowed us to capture the spectra of the sensor across all 15 Bragg wavelengths, ranging from 813 to 869 nm. We recorded the sensor's spectra at random curvatures and orientations (within a curvature range of 0.58–33.5 m$^{-1}$) while monitoring the reflective markers attached to the 30 cm long sensor using a motion capture system (Oqus 7+, Qualisys AB, Sweden). The data acquisition duration was 30 min for the $Test_1$ and 3 min for the $Test_2$ datasets. The acquisition rates in the FBG interrogator and the motion capture system were 75 and 200 Hz, respectively. The sensor's spectra and the coordinate values corresponding to its shape were synchronized with a tolerance of less than 3 ms.

In addition, we used a laser-cut curvature template (Fig. 5) to collect 320 samples for the $Test_3$ dataset, where specific regions of the sensor were intentionally bent. The curvature template features four grooves, enabling us to bend the sensor at the middle 30 mm area between the sensing planes 2 and 3, 3 and 4, 4 and 5, and 10 mm after the last sensing plane with a bending radius of 50 mm.

**Training setup**. The search space we defined for tuning the network's hyperparameters consists of the number of 1D

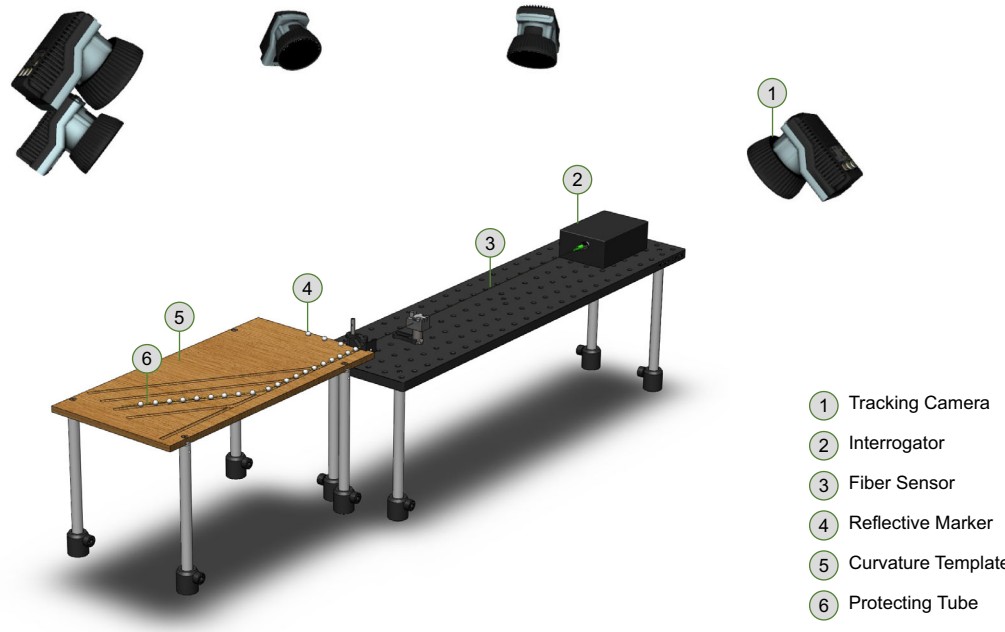

1  Tracking Camera
2  Interrogator
3  Fiber Sensor
4  Reflective Marker
5  Curvature Template
6  Protecting Tube

**Fig. 5 Experimental setup for data acquisition.** The motion capture system consisted of five tracking cameras (Oqus 7+, Qualisys AB, Sweden). For protection purposes, the fiber sensor was inserted in a Hytrel furcation tubing with an inner diameter of 425 μm and an outer diameter of 900 μm. Two v-clamps were used to hold the protection tubing securely and to fix the optical fiber in place before the insertion. Reflective markers with a diameter of 6.4 mm and an opening of 1 mm (X12Co., Ltd., Bulgaria) were affixed to the sensor. Additionally, a thermocouple was positioned near the sensor's base to monitor the temperature throughout the data acquisition process, ensuring that any sudden thermal fluctuations did not impact the sensor's signal.

**Table 2 The search criteria for hyperparameter optimization.**

| Hyperparameter | Search space | Selected values |
|---|---|---|
| Number of Conv1D layer | min: 1, max: 20, step: 1 | 5 |
| Number of FC layer | min: 1, max: 20, step: 1 | 5 |
| BN after each layer | true, false | – |
| Dropout after FC layer | true, false | – |
| Dropout rate | min: 0.1, max: 0.8 | – |
| Stride | min: 1, max: 2, step: 1 | – |
| Kernel size (max pooling layer) | min: 2, max: 3, step: 1 | – |
| Distribution of initial weights | standard, Xavier_uniform, Xavier_normal, Kaiming_uniform, Kaiming_normal | Xavier_normal |
| Learning rate | 0.01, 0.001, 0.0001, 0.00001 | 0.0001 |
| Sorting Conv1D layers | true, false | true |
| L2 regularization | 0.1, 0.01, 0.001, 0.0001, 0.00001, 0 | 0 |
| Threshold in SmoothL1 | any values between 0.0 and 5.0 | 4.04 |

*Conv1D 1D convolutional layer, FC fully connected layer, BN batch normalization.*

convolutional layers (Conv1D), the number of fully connected layers (FC), the layer settings, the choice of batch normalization (BN) and downsampling, training settings, and loss function parameters. The search criteria are outlined in Table 2.

In the designed network (Fig. 2), input samples with a batch size of 256 are first batch normalized and then fed into a Conv1D layer with 16 channels, followed by a max pooling layer with a kernel size of 3 and a stride of 2. The second Conv1D layer also has 16 channels, followed by a max pooling layer with a kernel size of 2. The third Conv1D layer has 32 channels, followed by a max pooling layer with a kernel size of 3 and a stride of 2. The fourth Conv1D layer also has 32 channels with a stride of 2, followed by a max pooling layer with a kernel size of 3. The last Conv1D layer has 256 channels, followed by batch normalization and a max pooling layer with a kernel size of 2 and a stride of 2. The extracted features are flattened to a 2048-long vector, fed into

5 FC layers, each with 2000 units. The first FC layer is followed by batch normalization, a dropout layer with a probability of 0.37, and two more FC layers. A batch normalization, an FC layer, a dropout layer with a probability of 0.16, and a fifth FC layer are the remaining layers before the final layer. The last layer is an FC layer that maps the output of the fifth FC layer into the target values, the relative coordinates. In all layers of this network architecture, the rectified linear unit (ReLU) serves as the activation function, and the kernel size for the Conv1D layers is 3. In this model, the Adam optimizer with a learning rate of 0.0001 minimizes the SmoothL1 loss function with a threshold of 4.04.

**Decoding the model's decisions.** Inspired by the concept of Gradient-weighted Class Activation Mapping (Grad-CAM), we

decoded the decisions made by our CNN (convolutional neural network)-based model. By decoding our model's decisions, we gained insights into which parts of the input spectra contribute to coordinate predictions. Grad-CAM is a widely used technique in image classification tasks that generates visual explanations from any CNN-based model without requiring re-training or architectural modifications. The gradient is a measure that shows the effect on the output caused by the input, indicating the part of the input with the highest impact on the model's output.

However, the gradient heat map produced by the last Conv1D layer has limited resolution due to the small output dimension in each channel. Therefore, instead of the gradient of the Conv1D layers, we computed the forward finite difference of the model's loss with respect to the input spectral elements. The spacing constant was chosen to be 0.1, higher than the spectral intensity noise level. In this method, we modified the intensity value of one spectral element and observed the resulting changes in the model's loss value. We repeated this process for all 190 spectral elements. The resulting color maps are illustrated in Figs. 3b and 4b, representing the impact of the changes in each spectral element on the model's SmoothL1 loss value. To analyze the contribution of each spectral element to the coordinate prediction of individual markers, we computed the Euclidean distance between the predicted coordinates of each marker before and after spectral modification. This allowed us to identify the spectral elements contributing to the relative coordinate prediction of each marker. By highlighting these spectral elements, we gained a better understanding of the factors influencing the model's predictions.

## Data availability
The datasets generated during and/or analyzed during the current study are available in the Academic Torrents repository.

## Code availability
The source code is available on GitHub.

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

## Acknowledgements

We gratefully acknowledge the funding of this work by Werner Siemens Foundation through the MIRACLE project. The authors express their appreciation to Yi Jiang for performing the eFBG calibration.

## Author contributions

All authors participated in the discussions and contributed to the completion of this paper. S.M.R. designed and built the experimental setup, conducted experiments, implemented the deep learning model, and, in collaboration with P.C.C., analyzed the results. C.S. implemented the hyperparameter optimization algorithm. M.A. and W.S. supplied the eFBG fiber sensor and validated the analytical MFD approach, serving as the baseline for sensor evaluation. S.F., G.R., and P.C.C. provided supervision throughout the entire research process. S.M.R. and P.C.C. wrote the paper with input from the other co-authors.

## Competing interests

The authors declare no competing interests.
