## [Peer Review File · Communications Engineering]

Reviewers' comments:

Reviewer #1 (Remarks to the Author):

Comment on the manuscript by S. M. Roodsari et al., "Deep-learning-based high spatial resolution fiber shape sensing":

I am struggling to make a decision on papers involved with deep learning. In my opinion, the authors described another application using an existing deep-learning method. They claimed an improved spatial resolution for fiber-optic shape sensing, but we do not know why. They have preliminary results on shape sensing and did not demonstrate real-world applications. For journals like Communications Engineering, the authors should have demonstrated an application that only the deep-learning sensors can do while the conventional fiber optic sensor not.

In Table 1, the MFD measurement errors are 111.3, 98.5, and 39.5 mm for a 30 cm long fiber, which is not at the current technological level. The resolution enhancement could be more convincing.

Considering the novelty and significance, I think the paper is more suitable for journals like Optics Express, or IEEE Journal of Lightwave Technology.

Reviewer #2 (Remarks to the Author):

The authors have presented their results on a study using machine learning for interpreting the output of a fibre Bragg grating (FBG) shape sensor. This builds on their previous work in using off-centred FBGs for shape sensing by directly relating the FBG changes to bending. Here they apply supervised machine learning, specifically, convolutional neural networks. I found the paper well written and the work rigorous and well thought out. It is hard to find much to fault with the work, so I recommend for publication and only make a few observations the authors may wish to consider to revise:

- In the introduction the authors write that sub millimetre distributed sensing require costly time or frequency domain analysis equipment, while FBGs only require cheaper spectral domain readout. However, frequency domain analysis is generally a type of spectral analysis. While I appreciate the type of equipment is different, on first glance these statements seem to be in conflict, so I suggest rewording.
- In Fig. 1, the spectra show obvious intensity changes when the fibre is perturbed, but how significant is the shift. I understand this is just an example, but the mechanism could be made more clear if they are either overlaid or perhaps vertical dashed lines so that the FBG shifts can be seen better.
- The "dictionary" test was quite interesting to me. I don't believe this is a standard test for machine learning applications, but I can see the validity. Particularly when the data is lab generated (c.f. something like Facebook or Google datasets), where there is risk the data is not independent. One thing that occurs to me is that this test is very similar to the use of k nearest neighbours, kNN (with $k = 1$), perhaps the authors could do a similar test but optimising for the value of k to thoroughly test that the deep learning is truly providing an advantage for their application.

Rebuttal letter accompanying the revised version of manuscript

Manuscript ID: COMMS-23-0226-T

“Deep-learning-based high spatial resolution fiber shape sensing”

*Samaneh Manavi Roodsari, Sara Freund, Martin Angelmahr, Georg Rauter, Wolfgang Schade, and
Philippe C. Cattin*

Communications Engineering

The authors express their gratitude to the referees for reviewing the paper and providing valuable feedback. Below, we address the comments point by point. In certain cases, our response to the comments has led to modifications in the paper, which are indicated in the submitted revision.

Reviewer 1, Comment 1

Comment on the manuscript by S. M. Roodsari et al., “Deep-learning-based high spatial resolution fiber shape sensing”:

I am struggling to make a decision on papers involved with deep learning. In my opinion, the authors described another application using an existing deep-learning method.

Response

We express our appreciation to the reviewer for dedicating their time to evaluate this manuscript. We also share their reservations concerning the use of deep-learning (DL) in fiber sensors. As physicists ourselves, we tried to model the bending-induced physical effects in optical fibers as accurate as possible, and developed special calibration tools to then estimate the shape of our eccentric FBG (eFBG) sensor. However, it became evident that the accuracy of shape estimation is confined to a few centimeters, which prompted our transition to the utilization of deep-learning as a substitute for our physical modeling approach.

The reason for selecting eFBGs for shape sensing is to develop an economically viable tracking solution that competes with commercially available fiber shape sensors. The key to achieving cost-effectiveness is substituting multicore fibers in commercial shape sensors with a single-mode fiber containing eFBG triplets. However, it’s important to note that the intensity-based nature of eFBG sensors presents a challenge in achieving precise shape estimation, even with high sensing density. Before opting for the DL technique to model our eFBG sensor, we conducted an in-depth investigation of the conventional analytical approach used for off-centered FBGs, which is based on measuring mode-field displacement (MFD).

Our findings indicate that DL models outperform the MFD approach, even without optimizing the sensing plane density. This is because DL models take the full spectrum of the eFBG sensor as input, including the Bragg resonance’s side slopes, to exploit the impact of bending-induced phenomena (*e.g.*, cladding mode coupling, bending-loss oscillations, and polarization-dependent losses) for accurately detecting complex deformations. In the forthcoming research, our objective is to learn more about the exact physical model and its associated parameters by studying the DL model we have trained.

Although we designed our DL model using existing optimization algorithms and layer functions from the TensorFlow library, it’s essential to acknowledge that fine-tuning the search criteria and layer settings is a problem-specific process. Particularly in regression tasks, pre-existing network architectures are not directly applicable, and a custom design based on the task’s constraints is needed.

Reviewer 1, Comment 2

The authors claimed an improved spatial resolution for fiber-optic shape sensing, but we do not know why.

Response

The enhancement of spatial resolution can be attributed to the DL model’s capability to accurately predict complex shapes, as showcased in the supplementary videos submitted with the manuscript. This includes its proficiency in detecting deformations occurring between sensing segments or after the last eFBG triplet, as illustrated in Figures 3 and 4 in the manuscript.

In the “Methods/Working principle of eFBG sensor” section on page 11 of the original manuscript, we presented three potential sources of information that the DL model utilizes for detecting deformations beyond the sensing segments of the fiber. These sources include cladding mode coupling in eFBGs, polarization-dependent loss in spectrometers, and bending loss oscillations in optical fibers, all of which are comprehensively explained in the provided references. However, conducting individual experiments to isolate and demonstrate the contribution of each of these effects to the spectral changes in deformed eFBG sensors is an intricate undertaking. Given that existing theories already substantiate the impact of such bending-induced effects, we decided not to pursue experimental demonstrations.

Reviewer 1, Comment 3

They have preliminary results on shape sensing and did not demonstrate real-world applications. For journals like Communications Engineering, the authors should have demonstrated an application that only the deep-learning sensors can do, while the conventional fiber optic sensor not.

Response

We thank the reviewer for bringing this valid point to our attention, and hope our response provides clarification on this matter.

In order to maintain brevity and generality in the Introduction section, we initially discussed the broad range of potential applications for the proposed shape sensing solution (lines 35 to 42). However, the primary inspiration behind the development of this tracking system originated from the need to navigate a flexible endoscope designed for minimally invasive laser osteotomy, as explained in the reference [1]. The envisioned flexible endoscope is 30 cm long and expected to undergo bending radii as tight as a few centimeters during surgical procedures. The ultimate objective is to achieve sub-millimeter accuracy in navigating the endoscope during surgery.

Conventional commercially available fiber shape sensors, such as distributed multicore fiber sensors interrogated by optical frequency domain reflectometers (OFDR), are constrained by their limited accuracy, which typically results in tip errors of up to 1% per unit length of the sensor. For a 30 cm long sensor, this equates to an approximate error of 3 mm. Consequently, such commercially available fiber shape sensors are unsuitable for the demanding requirements of this task.

Some examples of these limitations include challenges related to the signal-to-noise ratio in cross-correlation between OFDR traces, particularly at high spatial resolutions [4], and under high strain conditions [3, 6, 7]. These issues arise due to the dependence of the OFDR back-scattered trace patterns on the spatial distribution of inhomogeneities within the sensing fiber. Achieving sub-millimeter spatial resolution has proven to be a challenging endeavor, as cross-correlation tends to exhibit significant noise at resolutions other than 2.5 mm [2]. This led us to embark on the development of a novel shape sensing solution that overcomes the common limitations encountered by the state-of-the-art tracking systems.

Reviewer 1, Comment 4

In Table 1, the MFD measurement errors are 111.3, 98.5, and 39.5 mm for a 30 cm long fiber, which is not at the current technological level. The resolution enhancement could be more convincing.

Response

It's important to clarify that in Table 1, the eFBG sensor analyzed using the MFD method is not intended to represent the current state-of-the-art shape sensing technology. Instead, it serves as a baseline to illustrate what can be achieved through analytical methods, specifically MFD, when utilizing a low sensing density in eFBG single-core fiber sensors in combination with cost-effective interrogators. The primary objective is to demonstrate the significant advantages offered by the DL method when leveraging the entire eFBG spectral information that takes into account various bending-induced effects.

We extend our gratitude to the reviewer for bringing this to our attention and added this description to the caption of Table 1 for clarification.

While it is indeed feasible to improve shape reconstruction in the MFD method by increasing the sensing density (i.e., the number of eFBG triplets), it's worth noting, as mentioned in line 262 of the manuscript, that a total of 252 eFBGs would have been needed to attain the same level of accuracy provided by the DL model using only 15 eFBGs.

Corresponding section in the manuscript:

Optimum number of sensing planes. *A key factor in eFBG sensors when employing the MFD method is the number of sensing planes for detecting shape deformations. As with any other quasi-distributed shape sensor, the spacing between the sensing planes determines the sensor's spatial resolution in shape measurements. When dealing with complex shape deformations, a limited number of sensing planes (resulting in low spatial resolution) can lead to significant tip errors in methods that involve shape reconstruction (e.g., the MFD method). In this section, we present a theoretical analysis to determine the minimum number of sensing planes required in eFBG sensors when employing the MFD method, to achieve the same level of shape prediction accuracy as attained by our DL method using five sensing planes.*

In this theoretical analysis, we simulated the shape reconstruction error for different spatial resolutions. Our methodology involves interpolating the discrete curve points along the sensor's true shape, measured by the motion capture system, using a Spline with a resolution of 0.1 mm (this value was chosen empirically). Subsequently, we calculated the curvature and torsion—representing the curve's deviation from the osculating plane—at the query points. By utilizing the calculated curvatures and bending directions at the sensing planes, we reconstructed the spatial curve and compared it with the true shape.

For a 25 cm long sensor with 50 mm spatial resolution (equivalent to five sensing planes), the median tip error of the reconstructed shapes, evaluated using the $Test_1$ and $Test_2$ datasets, is approximately 50 mm. This error is nearly 16 times higher compared to the performance achieved by the DL approach (see Table 1). In order to achieve a median tip error of 3 mm, a similar spatial resolution is necessary, implying that the MFD method would require approximately 84 sensing planes consisting of 252 eFBGs.

Reviewer 2, Comment 1

The authors have presented their results on a study using machine learning for interpreting the output of a fibre Bragg grating (FBG) shape sensor. This builds on their previous work in using off-centred FBGs for shape sensing by directly relating the FBG changes to bending. Here they apply supervised machine learning, specifically, convolutional neural networks. I found the paper well written and the work rigorous and well-thought-out. It is hard to find much to fault with the work, so I recommend for publication and only make a few observations the authors may wish to consider to revise.

Response

We express our gratitude to the reviewer for dedicating their valuable time to assess this manuscript and for providing constructive feedback.

Reviewer 2, Comment 2

In the introduction, the authors write that sub millimetre distributed sensing require costly time or frequency domain analysis equipment, while FBGs only require cheaper spectral domain readout. However, frequency domain analysis is generally a type of spectral analysis. While I appreciate the type of equipment is different, on first glance these statements seem to be in conflict, so I suggest rewording.

Response

We extend our gratitude to the reviewer for bringing this to our attention, and have subsequently revised the relevant section in accordance with the feedback:

Accurate shape reconstruction necessitates high spatial resolution in off-axis strain measurement. With distributed fiber shape sensor, sub-millimeter spatial resolution can be achieved [4]. However, these sensors require the use of specialized and costly optical reflectometers to analyze the back scattered light and retrieve strain variations [20-23]. Moreover, the signal-to-noise ratio of the back scattering trace in such sensors depends on the spatial resolution and the level of applied strain. Quasi-distributed sensors, on the other hand, have more cost-effective readout unit systems (e.g., FBG interrogators). However, their spatial resolution is limited by the low sensing plane density [8,24], making them inapplicable for tracking complex shape deformations. Therefore, there is a need for a cost-effective, high-resolution, and accurate fiber shape sensing technique.

Reviewer 2, Comment 3

In Fig. 1, the spectra show obvious intensity changes when the fibre is perturbed, but how significant is the shift. I understand this is just an example, but the mechanism could be made more clear if they are either overlaid or perhaps vertical dashed lines so that the FBG shifts can be seen better.

Response

We added vertical dashed lines to Figure 1 and wish to express our thanks to the reviewer for suggesting this valid improvement.

It's important to highlight that the gratings in eFBG sensors are only $2\ \mu\text{m}$ away from the neutral axis, resulting in an expected Bragg wavelength shift within the range of a few pm (15 pm wavelength shift is reported in [5] for 5 mm bending radius). On the other hand, the eFBG spectra mainly undergo changes in intensity when the fiber is bent, depending on their angular and radial positions. Consequently, an intensity-based interrogation of eFBGs has been selected as it appears to be several orders of magnitude more sensitive than a wavelength interrogation of the identical sensor [5]. Therefore, in Figure 1, only alterations in the intensity ratio between eFBGs within the same sensing plane/segment are depicted.

The updated version of Figure 1 is presented on the following page.

Reviewer 2, Comment 4

The “dictionary” test was quite interesting to me. I don't believe this is a standard test for machine learning applications, but I can see the validity. Particularly when the data is lab generated (c.f. something like Facebook or Google datasets), where there is risk the data is not independent. One thing that occurs to me is that this test is very similar to the use of k nearest neighbours, kNN (with $k = 1$), perhaps the authors could do a similar test but optimising for the value of k to thoroughly test that the deep learning is truly providing an advantage for their application.

Response

The dictionary test is indeed similar to kNN, with a k value of 1 and yields equivalent error values. Nevertheless, it offers a slightly more intuitive means of illustrating the constraints associated with ground truth samples.

As per the recommendation, we proceeded to assess the performance of the kNN method with varying k values and compared the outcomes to those achieved using our developed DL approach.

Corresponding text:

To investigate this further, we compared the performance of our DL method with a dictionary-based algorithm. In this approach, a pre-defined dictionary was created using all training and validation samples. The shape prediction was then made by finding the closest spectrum to the test sample and presenting its corresponding shape. This technique is equivalent to k-nearest neighbors (kNN) algorithm with a k value of 1. The median tip errors for the Test₁ and Test₂ datasets using this dictionary-based algorithm are 5.9 mm and 50.0 mm with IQR values of 3.9 mm and 43.3 mm, respectively. We also evaluated the kNN algorithm with k values of 3, 5, 7, and 9 which resulted in median tip errors of 6.4, 7.8, 9.1, and 10.1 for the Test₁ dataset and 47.3, 46.1, 45.4, and 44.8 for the Test₂ dataset, respectively. All error values are higher compared to the errors obtained using our DL technique. This shows that our DL model generalizes well and provides more accurate shape predictions.

Figure 1: FBG configuration and working principle of the eFBG sensor.

References

- [1] MIRACLE Project. URL <https://dbe.unibas.ch/en/research/flagship-project-miracle/>.
- [2] M. F. Bado, J. R. Casas, A. Dey, C. G. Berrocal, G. Kaklauskas, I. Fernandez, and R. Rempling. Characterization of concrete shrinkage induced strains in internally-restrained rc structures by distributed optical fiber sensing. *Cement and Concrete Composites*, 120:104058, 2021.

- [3] M. Froggatt and J. Moore. High-spatial-resolution distributed strain measurement in optical fiber with rayleigh scatter. *Applied optics*, 37(10):1735–1740, 1998.
- [4] D. K. Gifford, B. J. Soller, M. S. Wolfe, and M. E. Froggatt. Distributed fiber-optic temperature sensing using rayleigh backscatter. In *2005 31st European Conference on Optical Communication, ECOC 2005*, volume 3, pages 511–512. IET, 2005.
- [5] C. Waltermann, K. Bethmann, A. Doering, Y. Jiang, A. L. Baumann, M. Angelmahr, and W. Schade. Multiple off-axis fiber bragg gratings for 3d shape sensing. *Applied optics*, 57(28):8125–8133, 2018.
- [6] C. Xu and Z. Sharif Khodaei. Shape sensing with rayleigh backscattering fibre optic sensor. *Sensors*, 20(14):4040, 2020.
- [7] S. Zhang, H. Liu, E. Darwish, K. M. Mosalam, and M. J. DeJong. Distributed fiber-optic strain sensing of an innovative reinforced concrete beam–column connection. *Sensors*, 22(10):3957, 2022.

REVIEWERS' COMMENTS:

Reviewer #1 (Remarks to the Author):

I have carefully read the revised manuscript and the rebuttal letter provided by the authors. I am pleased to see that the authors have taken my previous comments into consideration and made significant changes to the manuscript. Deep learning is a rapidly developing field with great potential for improving measurement accuracy in various applications, including fiber optic sensing. I believe that the proposed method in this manuscript has important implications for this field.

Therefore, I recommend this manuscript for publication in Communications Engineering.

Reviewer #2 (Remarks to the Author):

The authors have satisfactorily addressed the revisions that I suggested. I just note one minor correction, to add units for the errors listed in the sentence, "We also evaluated the kNN algorithm with k values of 3, 5, 7, and 9 which resulted in median tip errors of 6.4, 7.8, 9.1, and 10.1 for the Test1 dataset and 47.3, 46.1, 45.4, and 44.8 for the Test2 dataset, respectively."

Rebuttal letter accompanying the revised version of manuscript

Manuscript ID: COMMS-23-0226B

“Deep-learning-based approach for high spatial resolution fiber shape sensing”

*Samaneh Manavi Roodsari, Sara Freund, Martin Angelmahr, Carlo Seppi, Georg Rauter, Wolfgang Schade,
and Philippe C. Cattin*

Communications Engineering

The authors express their gratitude to the referees for reviewing the paper and providing valuable feedback.

Reviewer 1, Comment 1

I have carefully read the revised manuscript and the rebuttal letter provided by the authors. I am pleased to see that the authors have taken my previous comments into consideration and made significant changes to the manuscript. Deep learning is a rapidly developing field with great potential for improving measurement accuracy in various applications, including fiber optic sensing. I believe that the proposed method in this manuscript has important implications for this field.

Therefore, I recommend this manuscript for publication in Communications Engineering.

Response

We express our appreciation to the reviewer for dedicating their time to evaluate this manuscript. The reviewer's professional guidance contributes greatly to the advancement of our research. We look forward to the opportunity for our paper to contribute to the field.

Reviewer 2, Comment 1

The authors have satisfactorily addressed the revisions that I suggested. I just note one minor correction, to add units for the errors listed in the sentence, "We also evaluated the kNN algorithm with k values of 3, 5, 7, and 9 which resulted in median tip errors of 6.4, 7.8, 9.1, and 10.1 for the Test1 dataset and 47.3, 46.1, 45.4, and 44.8 for the Test2 dataset, respectively."

Response

We extend our gratitude to the reviewer for bringing this to our attention, and have subsequently revised the relevant section in accordance with the feedback. We express our appreciation to the reviewer for dedicating their time to evaluate this manuscript and providing constructive feedback.